# iGibson 2.0: Object-Centric Simulation for Robot Learning of Everyday Household Tasks

**Chengshu Li\*♠, Fei Xia\*♡, Roberto Martín-Martín\*♠**
**Michael Lingelbach♣, Sanjana Srivastava♠, Bokui Shen♠, Kent Vainio♠**
**Cem Gokmen♠, Gokul Dharan♠, Tanish Jain♠, Andrey Kurenkov♠**
**Karen Liu♠⋆, Hyowon Gweon◇⋆, Jiajun Wu♠⋆, Li Fei-Fei♠⋆, Silvio Savarese♠⋆**

Department of Computer Science♠, Electrical Engineering♡, Neurosciences IDP♣, Psychology◇
Institute for Human-Centered AI (HAI)⋆
Stanford University

**Abstract:** Recent research in embodied AI has been boosted by the use of simulation environments to develop and train robot learning approaches. However, the use of simulation has skewed the attention to tasks that only require what robotics simulators can simulate: motion and physical contact. We present iGibson 2.0, an open-source simulation environment that supports the simulation of a more diverse set of household tasks through three key innovations. First, iGibson 2.0 supports object states, including temperature, wetness level, cleanliness level, and toggled and sliced states, necessary to cover a wider range of tasks. Second, iGibson 2.0 implements a set of predicate logic functions that map the simulator states to logic states like `Cooked` or `Soaked`. Additionally, given a logic state, iGibson 2.0 can sample valid physical states that satisfy it. This functionality can generate potentially infinite instances of tasks with minimal effort from the users. The sampling mechanism allows our scenes to be more densely populated with small objects in semantically meaningful locations. Third, iGibson 2.0 includes a virtual reality (VR) interface to immerse humans in its scenes to collect demonstrations. As a result, we can collect demonstrations from humans on these new types of tasks, and use them for imitation learning. We evaluate the new capabilities of iGibson 2.0 to enable robot learning of novel tasks, in the hope of demonstrating the potential of this new simulator to support new research in embodied AI. iGibson 2.0 and its new dataset are publicly available at http://svl.stanford.edu/igibson/.

## 1 Introduction

Recent years, we have seen the emergence of many simulation environments for robotics and embodied AI research [1, 2, 3, 4, 5, 6, 7, 8, 9, 10]. The main function of these simulators is to compute the motion resulting from the physical contact-interaction between (rigid) bodies, as this is the main process that allows robots to navigate and manipulate the environment. This kinodynamic simulation is sufficient for pick-and-place and rearrangement tasks [11, 12, 13, 14]; however, as the field advances, researchers are taking on more diverse and complex tasks that cannot be performed in these simulators, e.g., household activities that involve changing the temperature of objects, their dirtiness and wetness levels. There is a need for new simulation environments that can maintain and update new types of object states to broaden the diversity of activities that can be studied.

We present iGibson 2.0, an open-source extension of the kinodynamic simulator iGibson with several novel functionalities. First and foremost, iGibson 2.0 maintains and updates new **extended physical states** resulting from the approximation of additional physical processes. These states include not only kinodynamics (pose, motion, forces), but also object's temperature, wetness level, cleanliness level, toggled and sliced state (functional states). These states have a direct effect on the appearance of the objects, captured by the high-quality virtual sensor signals rendered by the simulator.

Second, iGibson 2.0 provides a set of logical predicates that can be evaluated with a single object (e.g. `Cooked`) or a pair of objects (e.g. `InsideOf`). These logical predicates discriminate the continuous

---

*indicates equal contribution
correspondence to {chengshu,feixia,robertom}@stanford.edu

5th Conference on Robot Learning (CoRL 2021), London, UK.

physical state maintained by the simulator into semantically meaningful logical states (e.g. `Cooked` is `True` if the temperature is above a certain threshold). Complementary to the discriminative functions, iGibson 2.0 implements **generative functions** that sample valid simulated physical states based on logical states. Scene initialization can then be described as a set of logical states that the simulator can translate into valid physical instances. This enables faster prototyping and specification of scenes in iGibson 2.0, facilitating the training of embodied AI agents in diverse instances of the same tasks. We demonstrate the potential of this generative functionality with a new dataset of home scenes densely populated with small objects. We generate this new dataset by applying a set of hand-designed semantic-logic rules to the original scenes of iGibson 1.0.

Third, to facilitate the development of new embodied AI solutions to new tasks in these new scenes, iGibson 2.0 includes a **new virtual reality interface** (VR) compatible with the two main commercially available VR systems. All states are logged during execution and can be replayed deterministically in the simulator, enabling the generation *a posteriori* of additional virtual sensor signals or visualizations of the interactions and the development of imitation learning solutions.

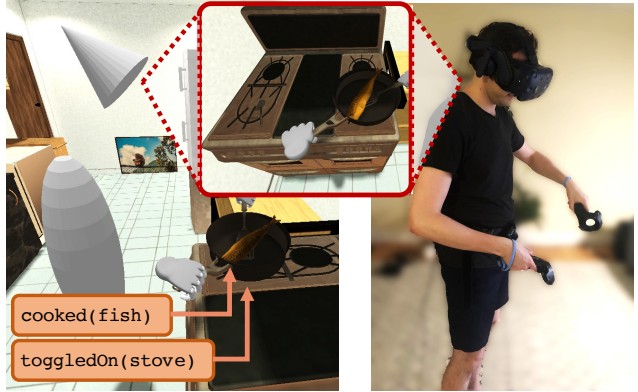

Figure 1: **Simulating new activities with iGibson 2.0** (*Left*) iGibson 2.0's simulates a set of extended physical states (temperature, functional state, cleanliness, wetness level) for objects, enabling studying and developing solutions to new household tasks such as cooking. The full physical state can be mapped to symbolic representation that facilitates sampling new instances of tasks. (*Right*) Humans can provide demonstrations for the new tasks with a novel virtual reality interface to enable policy learning.

We evaluate the new functionalities of iGibson 2.0 on six novel tasks for embodied AI agents, and apply state-of-the-art robot learning algorithms to solve them. These tasks were not possible before in iGibson or in alternative simulation environments. Additionally, we evaluate the use of the new iGibson 2.0 VR interface to collect human demonstrations to train an imitation learning policy for bimanual operations. While the previous version of iGibson and other simulators provide interfaces to control an agent with a keyboard and/or a mouse, these interfaces are insufficient for bimanual manipulation.

In summary, iGibson 2.0 presents the following contributions:

- A set of new physical properties, e.g. temperature, wetness and cleanliness level, maintained and updated by the simulator; and a set of unary and binary logical predicates that map simulated states to a logical state that have a direct connection to semantics and language,
- A set of generative functions associated with the logical predicates to sample valid simulated states from a given logical state, and a new rule-based mechanism exploiting these functions to populate the iGibson scenes with small objects placed at semantically meaningful locations to increase realism,
- A novel virtual reality interface that allows humans to collect demonstrations for robot learning,

We hope that iGibson 2.0 will open new avenues of research and development in embodied AI, enabling solutions to household activities that have been under-explored before.

## 2 Related Work

**Simulation environments with (mostly) kinodynamic simulation:** In the last years, the robotics and AI communities have presented several impressive simulation environments and benchmarks: iGibson [3, 1], Habitat AI [5], AI2Thor (and variants) [15, 6], ThreeDWorld [7], Sapien [4], Robosuite [9], VirtualHome [16], RLBench [8], MetaWorld [10], and more. They are based in physics engines such as (py)bullet [17], MuJoCo [18], and Nvidia Physx [19, 20], combined with rendering capabilities and usually enriched with a dataset of objects and/or scenes to use to develop embodied AI solutions. While these simulators have fueled research with new possibilities for training, testing and developing robotic solutions, they have skewed, with few exceptions, the exploration towards activities related to what they can simulate accurately: changes in kinematic states of (rigid or flexible) objects, i.e. Rearrangement tasks [11]. However, many everyday activities require the simulation

of other physical states that can be modified by the agents, like the temperature of objects and their level of wetness or cleanliness. Recent simulators have attempted realistic simulation of fluids and flexible materials [19, 21, 22, 23] or extended rigid body simulators to approximate the dynamics of soft materials [17, 24]. Compared with these simulators, iGibson 2.0 provides a simple but effective mechanism to simulate the temperature of objects that change based on proximity to heat sources. It also simulates fluids through a system of droplets that can be absorbed by objects and change their wetness level. Despite simpler than the accurate simulation of heat transfer, fluids and soft materials used by a few other simulators, iGibson 2.0 object-centric solution leads to realistic robotic behavior and motion in tasks involving changing temperature, handling liquids, or soaking objects.

**Simulation environments with object-centric representation:** In robotics, some simulators have adopted an object-centric representation with extended physical states, e.g. AI2Thor [15] and VirtualHome [16]. Both are based on Unity [25] and share a common set of functionalities. Actions in these simulators are predefined, discrete and symbolic, and characterized by preconditions ("what conditions need to be fulfilled for this action to be executable?") and postconditions ("what conditions will change after the execution of this action?"), similar to actions in the planning domain definition language (PDDL [26]) or STRIPS [27] but with the additional link to visual rendering of the predefined action execution and outcome. In AI2Thor and VirtualHome, some of the actions change the temperature of an object between two or more discrete values pre- and post-execution of an action, e.g. `raw` and `cooked`. This is fundamentally different to our approach in iGibson 2.0: instead of maintaining only a symbolic state (`raw`/`cooked`), we provide a simple simulation of the underlying physical process (e.g. heat transfer) leading to continuously varying values of temperature and other extended states. These states are then *mapped* into a symbolic representation through predicates (see Sec. 4). This provides a new level of detail in the execution of actions, where the agent can and should control the specific value of the object's extended states (temperature, wetness, cleanliness) to achieve a task, leading to more complex activities and more realistic execution. In the Appendix, we include a detailed comparison between iGibson 2.0 and other simulation environments in Table A.8.

**Simulation environments with virtual reality interfaces:** Researchers have used virtual reality interfaces before to develop robotic solutions with real robots [28, 29, 30]. VR has also been used in simulation environments to collect demos. VRKitchen [31] collected demos for five cooking tasks in simulation. While realistic looking, activities in VRKitchen are performed with primitive actions similar to the pre-condition/post-condition system of AI2Thor and VirtualHome, falling short of realistic motion. More physically realistic are the VR interactions with UnrealROX/RobotriX [32, 33] and ThreeDWorld [7], based on Unreal [34] and Nvidia Physx [20]. Our interface also enables realistic manipulation of objects, with additional features such as gaze tracking and assistive grasping to bridge the differences between simulation and the real-world.

## 3 Extended Physical States for Simulation of Everyday Household Tasks

To perform household tasks, an agent needs to change objects' states beyond their poses. iGibson 2.0 extends objects with five additional states: temperature, $T$, wetness level, $w$, cleanliness level (dustiness level, $d$, or stain level, $s$), toggled state, $TS$, and sliced state, $SS$. While some of these states could be different for different parts of an object, in iGibson 2.0 we simplify their simulation and adopt an **object-centric representation**: the simulator maintains a single value of each extended state for every simulated object (rigid, flexible, or articulated). This simplification is sufficient to simulate realistically household tasks such as cooking or cleaning. We assume that the extended properties are `latent`: agents are not able to observe them directly. Therefore, iGibson 2.0 implements a mechanism to change objects' appearance based on their latent extended states (see Fig. 2), so that visually-guided agents can infer the latent states from sensor signals.

We further impose in iGibson 2.0 that every simulated object should be an instance of an existing object category in WordNet [35]. This semantic structure allows us to associate characteristics to all instances of the same category [36, 37]. For example, we further simplify the simulation of extended states by annotating what extended states each category need. Not all object categories need all five extended states (e.g., the temperature is not necessary/relevant for non-food categories for most tasks of interest). The extended states required by each object category are determined by a crowdsourced annotation procedure in the WordNet hierarchy.

In the following, we explain the details of the five extended states and the way they are updated in iGibson 2.0. For a full list of object states (kinematics and extended), see Table A.4.

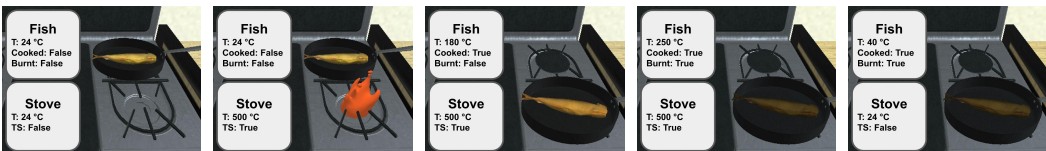

(a) **Object temperature:** *(From left to right)* A stove –toggleable heat source by proximity– is toggled on (second from left), and starts to heat a nearby fish. The temperature and appearance of the fish changes due to proximity to the heat source reaching the temperature to be cooked (third from left). Additional heat elevates further the fish's temperature, burning it (fourth from left). After the stove is toggled off, the fish changes back to room temperature, keeping the appearance that corresponds to the maximum temperature reached (most right). The temperature system with heat sources and sinks, enable visually guided execution of household activities such as cooking.

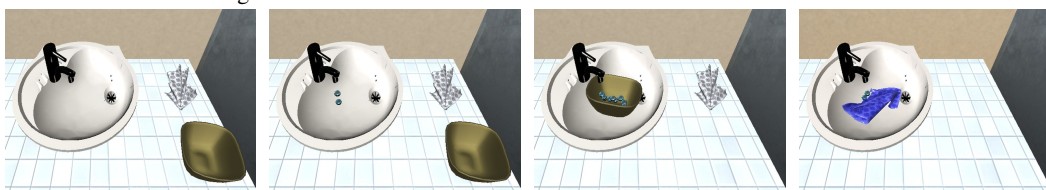

(b) **Object wetness level:** *(From left to right)* A sink –toggleable droplet source– is toggled on, and starts to create droplets (second from left). Droplets can be contained in a receptacle to be poured on other objects (third from left). An object that can change its wetness level gets in contact with the droplets and absorbs them, changing its appearance (most right). With the droplets system, iGibson v2.0 provides a simplified liquid simulation sufficient to perform common household activities.

Figure 2: Extended object states I): temperature and wetness level

**Temperature:** To update temperature, iGibson 2.0 needs to approximate the dynamics of heat transfer from heat sources (hot) or sinks (cold). To that end, we annotate object categories in the WordNet hierarchy as `heat sources` or `heat sinks`. Heat sources elevate the objects' temperature over room temperature ($23°$) towards the source's heating temperature, also annotated per category; similarly, heat sinks decrease the temperature under room temperature towards the sink's cooling temperature. The rate the objects change their temperature towards the temperature of the source/sink is a parameter annotated per category with units $°\,\text{C/s}$. We consider two types of heat source/sink: source/sinks that change the temperature of objects if they are proximal enough (e.g., a stove) and source/sinks that change the temperature of objects if they are inside of them (e.g., a fridge or an oven). Additionally, some of the sources/sinks need to be toggled on (see Functional State below) before they can change the temperature of other objects (e.g., a microwave). For each object that can change temperature, the simulator first evaluates if it fulfills the conditions to be heated or cooled by a heat source/sink. Assuming that an object with temperature $T_o$ fulfills the conditions of a source/sink with heating/cooling temperature $T$ and changing rate $r$, the temperature of the object at each step is updated by $T_o^{t+1} = T_o^t \left(1 + \Delta_{sim} \times r \times (T - T_o^t)\right)$, where $\Delta_{sim}$ is the simulated time.

iGibson 2.0 maintains also a historical value of the `maximum temperature` that each object has reached in the past, $T_o^{\max} = \max T_o^t$ for $t \in [0, \ldots, t_{\text{now}}]$. This value dictates the appearance of an object: if the object reached cooking or burning temperature in the past, it will look cooked or burned, even if its current temperature is low. Fig. 2a depicts the temperature system in action.

**Wetness Level:** Similar to temperature, iGibson 2.0 maintains the level of wetness for each object that can get `soaked`. This level corresponds to the number of *droplets* that have been absorbed by the object. In iGibson 2.0, the system of droplets approximates liquid/fluid simulation. Specifically, droplets are small particles of liquid that are created in droplet sources (e.g. faucets), destroyed by droplet sinks (e.g. sinks), and absorbed by soakable objects (e.g. towels). They can also be contained in receptacles (e.g. cups) and poured later, leading to realistic behavior for the simulation of several household activities involving liquids, illustrated in Fig. 2b.

**Cleanliness – Dustiness and Stain Level:** A common task for robots in homes and offices is to clean dirt. This dirt commonly appears in the form of dust or stains. In iGibson 2.0, the main difference between dust and stains is the way they get cleaned: while dust can be cleaned with a dry cleaning tool like a cleaning cloth, stains can only be cleaned with a soaked cleaning tool like a scrubber. To clean a particle of dirt (dust or stain), the right part of a cleaning tool should get in physical contact with the particle. Once a dirt particle is cleaned, it disappears from objects' surface.

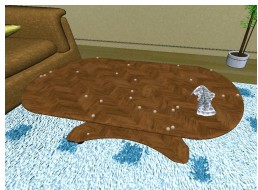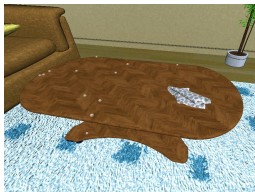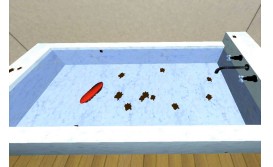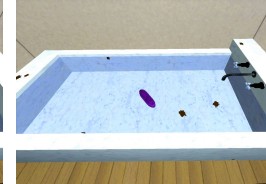

(a) **Object cleanliness level:** *(From left to right)* An object is initialized with dust particles that can be cleaned with a cloth; Another object is initialized with stains that require a wet tool (a scrubber) to be removed; Realistic behaviors and motion are required in iGibson v2.0 to change the cleanliness level of the objects.

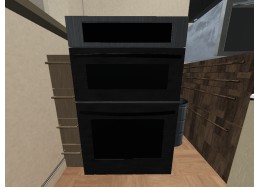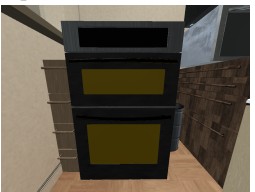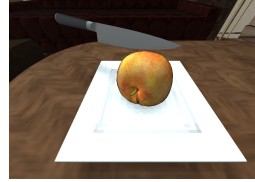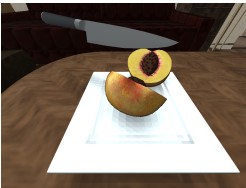

(b) **Object toggling state:** An object with the toggled state, an oven, is initially off. When it is toggled on, its appearance changes indicating the transition.

(c) **Object slice state:** An object with the sliced state, a peach, is sliced after exerting enough force with the right part of a slicing tool (the sharp edge of a knife)

Figure 3: Extended object states II): cleanliness level (dustiness, stains), toggling, slice state

In iGibson 2.0, objects can be initialized with visible dust or stain particles on its surface. The number of particles at initialization corresponds to a 100% level of dustiness, $d$, or stains, $s$, as we assume that dust/stain particles cannot be generated after initialization. As particles are cleaned, the level decreases proportionally to the number of particles removed, reaching a level of 0% dustiness, $d$, or stains when the object is completely clean (no particles left). This extended state allows simulating multiple cleaning tasks in our simulator: the agent needs to exhibit a behavior (motion, use of tools) similar to the one necessary in the real world. Fig. 3a depicts an example of the cleanliness level simulation in iGibson 2.0, for both dust and stain particles.

**Toggled State:** Some object categories in iGibson 2.0 can be toggled on and off. iGibson 2.0 maintains and updates an internal binary functional state for those objects. The functional state can affect the appearance of an object, but also activate/deactivate other processes, e.g. heating food inside a microwave requires the microwave to be toggled on. To toggle the object, a certain area needs to be touched. For object models of categories that can be toggled on/off, we annotate a `TogglingLink`, an additional virtual fixed link that needs to be touched by the agent to change the toggled state. Fig. 3b depicts an example of an object that can change its toggled state, an oven.

**Sliced State:** Many cooking activities require the agent to slice objects, e.g. food items. Slicing is challenging in simulation environments where objects are assumed to have a fixed (rigid or flexible) 3D structure of vertices and faces. To approximate the effect of slicing, iGibson 2.0 maintains and updates a sliced state in instances of object categories that are annotated as `sliceable`. When the sliced state transitions to True, the simulator replaces the whole object with two halves. The two halves will be placed at the same location and inherit the extended states from the whole object (e.g. temperature). The transition is not reversible: the object will remain sliced for all upcoming simulated time steps. Objects can only be sliced into two halves, with no further division. The sliced state changes when the object is contacted with enough force (over a slicing force threshold for the object) by a slicing tool, e.g. a knife. Objects of these categories are annotated as `SlicingTool`. If an object is a slicing tool, it will undergo a second annotation process to obtain a new virtual fixed link that acts as `SlicingLink`, the part of the slicing tool that can slice an object, e.g., the sharp edge of a knife. Fig. 3c depicts an example of a peach being sliced by a knife. For more information about update rules for all extended physical states, please refer to see Sec. A.2.

## 4 Logical Representation of Physical States

The new extended object states from iGibson 2.0 are sufficient to simulate a new set of household activities in indoor environments. However, there is a semantic gap between the extended states (e.g. temperature or wetness level) and the natural description of activities in a household setup (e.g. cooking apples). To bridge this gap, we define a set of functions that map the extended object states to logical states for single objects and pairs of objects. The logical states are semantically grounded on common natural language representing properties such as `cooked` or `dusty`.

The list of logical predicates covers kinematic states between pair of objects (`InsideOf`, `OnTopOf`, `NextTo`, `InContactWith`, `Under`, `OnFloor`), states related to the internal degrees of freedom of articulated objects (`Open`), states based on the object temperature (`Cooked`, `Burnt`, `Frozen`), wetness and cleanliness level (`Soaked`, `Dusty`, `Stained`), and functional state (`ToggledOn`, `Sliced`). A complete list of the logic predicates with detailed explanation is included in Table A.1. They allow iGibson 2.0 to map a physical simulated state into an corresponding logical state.

## 4.1 Generative System based on Logical Predicates

Logical predicates map multiple physically simulated states to the same logical state, e.g., all relative poses between to objects that correspond to being `onTop`. In addition to this discriminative role, we include functionalities in iGibson 2.0 to use logical predicates in a generative manner, to describe initial states symbolically that can be used to initialize the simulator. iGibson 2.0 includes a sampling mechanism to create valid instances of tasks described with logical predicates. This mechanism facilitates the creation of multiple semantically meaningful initial states, without the laborious process of manually annotating the initial distributions per scene.

The process of sampling valid object states is different depending on the nature of the logical predicate. For predicates based on objects' extended states such as `Cooked`, `Frozen` or `ToggledOn`, we just sample values of the extended states that satisfy the predicate's requirements, e.g., a temperature below the annotated freezing point of an object to fullfil the predicate `Frozen`. Particles for `Dusty` and `Stained` are sampled on the surface of an object following a pseudo-random procedure. Generating initial states to fulfill kinematic predicates such as `OnTopOf` or `Inside` is a more complex procedure as the underlying physical state (the object pose) must lead to a stationary state (e.g., not falling) that does not cause penetration between objects. Each kinematic predicate is implemented differently, combining mechanisms that include ray-casting and analytical methods to verify the validity of sampled poses. For example, to sample a state that satisfies `Inside(A, B)`, we implement a procedure that generates 6D poses inside of object `B` and we evaluate that 1) a bounding box of the the size of object `A` does not penetrate object `B` and rays cast from `A` intersect object `B` from evaluated by casting rays from the the internal box. For more details on the sampling mechanism, see Sec. A.2.

## 4.2 iGibson 2.0 Scenes with Realistic Object Distribution Created by Generative System

One common issue that limits the realism of indoor scenes in simulation is that they are less densely populated than those in the real world. Creating highly populated simulated houses is usually a laborious process that requires manually selecting and placing models of small objects in different locations. Thanks to the generative system in iGibson 2.0, this process can be extremely simplified. The users only need to specify a list of logical predicates that represent a realistic distribution of objects in a house.

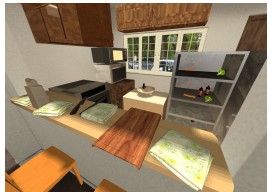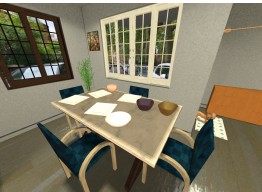

Figure 4: **Densely populated ecological scenes from the iGibson 2.0 dataset:** Objects are sampled in semantically meaningful locations using a set of semantic rules and the generative system based on logical statements, increasing realism for robot learning and VR experience.

We provide as part of iGibson 2.0 a set of semantic rules to generate more populated scenes, and a new version of the iGibson original 15 fully interactive scenes, populated with additional small objects as a result of the application of the rules. Given an indoor scene with multiple types of rooms (`kitchen`, `bathroom`, ...) that are populated with furniture containers and appliances (`fridge`, `cabinet`, ...), the semantic rules define the probabilities for object instances of diverse categories to be sampled in certain container type in a given room type, e.g. $p(\texttt{InsideOf(Beer, Fridge)} \wedge \texttt{InsideOf(Fridge, Kitchen)})$. To generate the more densely populated versions of the 15 scenes, we first collect a large number of small 3D object models created by artists and we annotate them with semantic categories (e.g. `cereal`, `apple`, `bowl`) and realistic dimensions. Then, we apply in a random sequence the logic predicates using the generative system explained in Sec. 4.1, increasing the number of objects in the scene by over 100. The result is depicted in Fig. 4.

## 5 Virtual Reality Interface

iGibson 2.0's new functionalities enable modeling new household activities and generating multiple instances in more densely populated scenes. To facilitate research in these new, complex tasks,

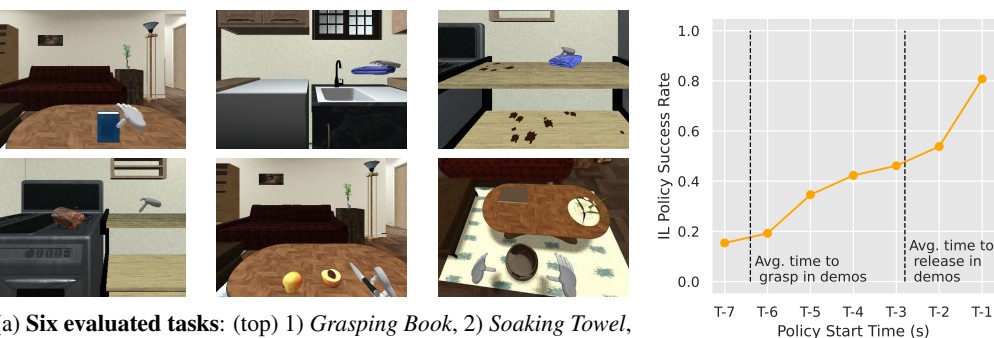

(a) **Six evaluated tasks**: (top) 1) *Grasping Book*, 2) *Soaking Towel*, 3) *Cleaning Stained Shelf*; (bottom) 4) *Cooking Meat*, 5) *Slicing fruit*, 6) *Bimanual Pick and Place*.

(b) **Success rate of IL in** *Bimanual Pick and Place*

Figure 5: **Evaluation:** We create six novel household tasks in iGibson 2.0 where the agents have to manipulate extended states. These tasks cannot be studied in other simulation environments.

iGibson 2.0 includes a novel virtual reality (VR) interface compatible with major commercially available VR headset through OpenVR [38]. One of the goals is to allow researchers to collect human demonstrations, and use them to develop new solutions via imitation (see Sec. 6).

iGibson 2.0's VR interface creates an immersive experience: humans embody an avatar in the same scene and for the same task as the AI agents. The virtual reality avatar (see Fig. 1) is composed of a main body, two hands and a head. The human controls the motion of the head and the two hands via the VR headset and hand controllers with an optional, additional tracker for control of the main body. Humans receive stereo images as generated from the point of view of the head of the virtual avatar, at at least 30 fps (up to 90 fps) using the PBR rendering functionalities of iGibson [1].

**Grasping in VR:** Grasping in the real-world, while natural to adult humans, is a complex experience that proves difficult to reproduce with virtual reality controllers. Our empirical observations revealed that while using solely physical grasping, user dexterity was significantly impaired relative to the real-world resulting in unnatural behavior when manipulating in VR. To provide a more natural grasping experience, we implement an assistive grasp (AG) mechanism that enables an additional constraint between the palm and a target object after the user passes a grasp threshold (50% actuation) and provided the object is in contact with the hand, between the fingers and the palm. This facilitates grasping of small objects, and prevents object slippage. To not render grasping artificially trivial, the AG connection can break if the constraint is violated beyond a set threshold, such as while lifting heavy objects or during intense acceleration, encouraging natural task execution that leverages careful motions and bimanual manipulation. Please refer to Sec. A.1 for additional details about AG.

**Navigating in VR:** Navigation of the avatar is controlled by the locomotion of the human. However, the VR space is much smaller than the typical size of iGibson 2.0 scenes. To navigate between rooms, we configured a touchpad in the hand controller that humans can use to translate the avatar.

## 6 Evaluation

In our evaluation, we test the new functionalities of iGibson 2.0 explained above and that sets it apart from other existing environments. First, we create a set of tasks that showcase the new extended states (see Fig. 5a) as their modification is required to achieve the tasks. In our experiments, we make use of our discriminative and generative logical engine to detect task completion and create multiple instances of each task for training. Then, we use our novel VR interface to collect human demonstrations to train an imitation learning policy for a bimanual task.

**Experimental Setup:** To evaluate and demonstrate the new capabilities of iGibson 2.0, we create the following six novel tasks for embodied AI agents: **1)** *Grasping Book:* the agent has to grasp a book `OnTopOf` a table and lift it. This task demonstrates the kinematic logical states; **2)** *Soaking Towel:* the agent has to soak a cleaning tool (a towel) with water droplets from a sink. This task demonstrates the liquid/droplets system. **3)** *Cleaning Stained Shelf:* the agent has to clean a stained shelf using a cleaning tool. This task showcases the capabilities to update the cleanliness level of an object. **4)** *Cooking Meat:* the agent has to cook a piece of meat by placing it on a heat source (a burning stove) and waiting for enough time for the temperature to rise. This task showcases the capabilities to update the temperature of an object. **5)** *Slicing Fruit:* the agent has to slice a fruit by exerting enough force with the `SlicingLink` of a slicing tool (a knife). This task showcases

the capabilities to simulate sliceable objects and to model the interaction between cutting tools and sliceable objects. **6)** *Bimanual Pick and Place:* The agent has to pick up a heavy object (a cauldron) and place it `OnTopOf` a table. Manipulating this object requires bimanual interaction as its mass is $50\,\mathrm{kg}$ and each hand can only exert at most $300\,\mathrm{N}$. Demonstrating bimanual manipulation is natural and easy with our novel VR interface, which is not the case with previous interfaces such as keyboard [1].

We conduct two sets of experiments to evaluate the current robot learning algorithms on these tasks. First, we train agents with a state-of-the-art reinforcement learning (RL) algorithm, Soft-Actor Critic (SAC [39]). For embodiment, we use a bimanual humanoid robot (the one used in VR) and a Fetch robot. The agent receives RGB-D images from its onboard sensors and proprioception information as observation and outputs the desired linear and angular velocities of the right hand (assuming the rest of the agent is stationary). We adopt the "sticky mitten" simplification [11] for grasping that creates a fixed constraint between the hand and the object when they get into contact. We also conduct experiments that remove such simplification for the Fetch robot, in which case the action space includes one additional DoF for grasping. Second, we train agents with a standard imitation learning (IL) algorithm, behavioral cloning [40] on *Bimanual Pick and Place*. We collected 30 demonstrations, more than 6500 frames in total ($\sim$215 s). The agent receives ground-truth states of the objects and proprioception information as observation, and outputs the desired linear and angular velocities for both hands. Additional information about the experimental setup can be found in Sec. A.3.

**RL experiments:** With the simplified grasping mechanism, the agents trained with SAC for both the bimanual humanoid and the Fetch embodiment achieve $100\%$ success rate for *Grasping Book*, *Soaking Towel*, *Cleaning Stained Shelf*, *Cooking Meat* tasks. For *Slicing Fruit*, the agents achieve only $15\%$ and $0\%$ for bimanual humanoid and Fetch robot respectively, due to the increased accuracy necessary to align the knife blade with the fruit. The agent using bimanual humanoid achieves $0\%$ success in the *Bimanual Pick and Place* task because of the difficulties of controlling and coordinating both hands. Additionally, we evaluate the performance with the Fetch robot in more realistic conditions, without any simplification for grasping, and observe a significant drop in performance, achieving $25\%$ success rate for 2 tasks and $0\%$ for the other 3 (see Fig. A.2 in the Appendix). This indicates that successful grasping for diverse objects is a significant challenge in these manipulation tasks. To test generalization, we conducted an ablation study in which we train policies with three different levels of variability in *Soaking Towel* –no variations, different poses, different objects and poses– and evaluate them on an unseen setup (an unseen object with randomized initial pose). The policies achieve success rate of $19\%$, $79\%$, and $87\%$, respectively. This study shows that it's essential to train with diverse object models and initial states to obtain robust policies, and the generative system of iGibson 2.0 facilitates it by specifying a few logical states that describe the initial scene. The attached video shows the policies performing the tasks trained using iGibson 2.0's new extended physical states and logical predicates that help generating task instances and discriminating their completion.

**IL experiments in *Bimanual Pick and Place* with VR demonstrations:** The trained policy in the full task diverges and fails, even after training with 30 demonstrations. We hypothesize that the agent suffers from covariate shift [41], visiting state space not covered by demonstrations, due to the different strategies demonstrated in VR for this long task (300+ steps). We then evaluate if the policy can successfully perform the last part of the task. We take a successful human demonstration and initialize a few seconds before task completion. We then query the IL policy from this point onward to control the agent. The results of this experiment are shown in Fig 5b. The policy achieves $19\%$ and $46\%$ success rate when starting $6\,\mathrm{s}$ and $3\,\mathrm{s}$ away from the goal. This experiment demonstrates the potential of the new VR interface of iGibson 2.0 for generating demonstrations for IL. We believe that our new interface will open new avenues for research in bimanual manipulation, hand-eye coordination, and coordination of robot base and robot arm.

## 7 Conclusion

We presented iGibson 2.0, an open-source simulation environment for household tasks with several key novel features: 1) an object-centric representation and extended object states (e.g. temperature, wetness and cleanliness level), 2) logical predicates mapping simulation states to logical states, and generative mechanism to create simulated worlds based on a given logical description, and 3) a virtual reality interface to easily collect human demonstrations for imitation. We demonstrate in multiple experiments the new avenues for research enabled by iGibson 2.0. We hope iGibson 2.0 becomes a useful tool for the community, and facilitates the development of novel embodied AI solutions.

## Acknowledgments

This work is in part supported by ARMY MURI grant W911NF-15-1-0479 and Stanford Institute for Human-Centered AI (SUHAI). S. S. is supported by the National Science Foundation Graduate Research Fellowship Program (NSF GRFP) and Department of Navy award (N00014-16-1-2127) issued by the Office of Naval Research. S. S. and C. L. are supported by SUHAI Award # 202521. R. M-M is supported by SAIL TRI Center – Award # S-2018-28-Savarese-Robot-Learn. F. X. and B. S. are supported by Qualcomm Innovation Fellowship. M. L. is supported by Regina Casper Stanford Graduate Fellowship.

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
