# OpenReview forum: "iGibson 2.0: Object-Centric Simulation for Robot Learning of Everyday Household Tasks"
_robot-learning.org/CoRL/2021/Conference — CoRL2021 Poster_

### Official Review · Reviewer_adeL · 2021-07-23

**Originality:** Good
**Technical Quality:** Good
**Clarity Of Presentation:** Good
**Impact:** 2

**Recommendation:**

Weak Accept: I recommend accepting the paper, but will not argue for my recommendation if the majority of other reviewers have a different opinion.

**Summary:**

The paper proposes the iGibson 2.0 simulation framework, which when compared to prior work, focuses on tasks involving object properties (material, wetness, temperature, etc.) and logical states (raw, cooked). The authors also provide a VR interface for users to interact with the environment - the demonstrations could be collected for the purpose of imitation learning. Six novel tasks were proposed (grasping a book from a table, cooking meat, slicing fruit, etc.) which are unique to the iGibson 2.0 simulator and imitation learning results were shown on the grasping task.

**Issues:**

I have some further questions/comments/feedback:

I would suggest showing clear comparisons, in the form of a table, to illustrate the differences between iGibson 2.0 and prior work (since there are a lot of simulators).

Some miscellaneous questions on the implementation details (it would be great to include them in the paper)
- How is the sliced peach mesh represented?
- How do you place the fish in the pan at the right stable pose? Is this hardcoded?
- Can the fish be burnt, do you model that? (instead of being in the raw vs cooked state)
- How many assets are there in total? For instance, is there only 1 peach? There seems to be very limited assets in general
- Do you have licenses for the assets when they are publicly released?

**Reviewer Expertise:**

Excellent: Expert knowledge on the topic of the paper

**Strengths And Weaknesses:**

iGibson 2.0 is a really impressive simulator that is needed for the next step in embodied AI research to reason about more complex semantic tasks.

My biggest concern for this work is that it is not grounded in a solid application in robotics. How could the policies or demonstrations collected in this simulator be relevant for a real robot? If there are some sim2real transfer capabilities that would be a more valid contribution for CoRL and the robotics community. Otherwise, this work is more suited for a graphics or VR conference.

The grasping demo is not a contribution since there are other more established methods of doing sim2real grasping (DexNet, 6 DOF GraspNet, etc.). The other tasks look interesting.

**Summary Of Recommendation:**

The submission in its current form is a weak accept.

---

> ### Author Response · Authors · 2021-08-27
> **Response to reviewer adeL 1/2**
>
> We thank the reviewer for the constructive feedback and we provide responses as below:
>
>
>
> > I would suggest showing clear comparisons, in the form of a table, to illustrate the differences between iGibson 2.0 and prior work (since there are a lot of simulators).
>
> We included a comprehensive comparison table of different simulation environments in Appendix A.5.
>
>
> > Had there been more engineering/systems contributions or robotics applications, I would have recommended to accept it.
>
> We address this concern in two parts. First, we provide more insights into engineering/system contributions with the simulation environment. There has been an very significant amount of engineering contributions in this work:
>
> - Object centric representation (semantic attributes, e.g. Cookable, Soakable, and physical attributes, e.g. mass, center of mass)
> - Extended object states and proximity and contact-based update functions (e.g. wetness, cleanliness, temperature)
> - Ray-casting and proximity-based checking functions for all logical states
> - Ray-casting based sampling functions for all logical states
> - Virtual reality interface (OpenVR integration, an assistive grasping mechanism that controls a realistic hand with 1DoF trigger, an intuitive interface that allows the agent to do navigation and manipulation at the same time, see Appendix A.1)
> - Performance improvement in the physics engine. We improve on several fronts compared with previous works 1) object loading 2) the object sleeping mechanism of pybullet 3)  joining fixed links, 4) lazy update of object poses in renderer (see Appendix A.4)
>
> Second, we provide clarification about robotics applications. Although our work has focused primarily on simulation, our goal is to use simulation as a proxy to develop solutions to real-world robotics problems. All of our experiments are inherently robotic applications: pick and place, wiping, slicing, cooking, etc.
>
> To alleviate the concern of unrealistic embodiment, we will additionally include results of RL experiments with the Fetch robot embodiment and share our findings in Appendix A.3 by the rebuttal deadline. iGibson 2.0 also includes models of other real robots such as Freight, LoCoBot, Turtlebot, Quadcopter, etc.
>
> With respect to sim2real transferability, previous work [1] using the Fetch robot in iGibson has shown that the virtual sensor simulation is very realistic (see page 16) and such a platform has the potential to develop solutions that transfer to the real world. Similar findings have been confirmed by other researchers using other types of robots for navigation tasks, e.g. four-wheeled robots [2] and legged robots [3].
>
> We also want to note that transferring mobile manipulation (MM) policies from simulation to the real world is an open research problem. While there have been successful transfer of navigation and stationary manipulation policies, the community is still a few steps away from transferring MM policies to the real world. Nonetheless, it is our plan to test some of our experiments on real robots in the near future now that we have regained access to the laboratory.
>
> ### References
>
> - [1] Xia, F., Li, C., Martín-Martín, R., Litany, O., Toshev, A., & Savarese, S. (2020). ReLMoGen: Leveraging motion generation in reinforcement learning for mobile manipulation. arXiv preprint arXiv:2008.07792.
> - [2] Meng, X., Xiang, Y., & Fox, D. (2021). Learning Composable Behavior Embeddings for Long-Horizon Visual Navigation. IEEE Robotics and Automation Letters, 6(2), 3128-3135.
> - [3] Li, T., Calandra, R., Pathak, D., Tian, Y., Meier, F., & Rai, A. (2021). Planning in learned latent action spaces for generalizable legged locomotion. IEEE Robotics and Automation Letters, 6(2), 2682-2689.

---

> > ### Comment · Reviewer_adeL · 2021-09-03
> > **Thanks for the response**
> >
> > Hi authors, thanks for the response, adding the table and demonstrating the Fetch grasping experiments. Thanks for the clarifications as well. I have updated my score.
> >
> > I think there will be broader impact if the assets are licensed to be used outside iGibson 2.0 (I'm assuming its not possible if they are bought from a third party).

---

> ### Author Response · Authors · 2021-08-27
> **Response to reviewer adeL 2/2**
>
> > How is the sliced peach mesh represented?
>
> A sliceable object is represented as an URDF (the whole object) that is replaced by two URDFs (the object parts) when the object is interacted in the right way, e.g., with a sharp tool applying enough force on its surface. Specifically, when the sliced state transitions to `True`, the simulator replaces the whole object with two object parts. The two parts will inherit the extended states from the whole object (e.g. temperature) onwards.
>
> > How do you place the fish in the pan at the right stable pose? Is this hardcoded?
>
> No, it is not hardcoded. To spawn objects, we use the sampling functions of kinematic logical states (e.g. `onTop`, `inside` - see Appendix A.2 for more details), and then let the objects fall due to gravity until they reach a stable pose after a few steps of physics simulation.
>
> > Can the fish be burnt, do you model that? (instead of being in the raw vs cooked state)
>
> Yes, the fish can burn. We model that for all cookable objects. Depending on the value of the continuous temperature variable, the following logic predicates will return a boolean value: Frozen, Cooked, and Burnt.
>
> > “How many assets are there in total? For instance, is there only 1 peach? There seems to be very limited assets in general. Do you have licenses for the assets when they are publicly released?”
>
> The assets are part of a parallel effort BEHAVIOR, a benchmark for 100 household activities. It includes a dataset of 1217 objects across 391 categories, ranging from food items to tableware, from home decorations to office supplies, and from apparel to cleaning tools. On average, each object category has approximately 3 different models. Our license agreement allows our users to use these assets within iGibson 2.0. We are committed to publicly release the assets once the review process is over. We also would like to highlight that iGibson 2.0 allows users to add new object models and categories of their choice easily.

---

> ### Author Response · Authors · 2021-08-31
> **Follow-up response**
>
> Dear Reviewer,
>
> We would like to thank you again for your time and feedback on iGibson 2.0.
>
> Since we posted our response last Thursday, we have made additional progress towards addressing the raised concerns.
>
> First, to address the concern of applicability to robotics and realism of embodiment, we conducted additional RL experiments with the Fetch robot, with simplified grasping and without it. When trained with simplified grasping, the RL policy achieves perfect success for 4 out of the 5 tasks (similar to bimanual humanoid) while Slicing Fruit task remains unsolved (because it requires a precise alignment between the knife blade and the fruit). All the joint actuations of Fetch are realistically simulated in Bullet, except the grasping. When trained without simplified grasping, in which case the robot relies on the friction between the gripper fingers and the object to grasp it, the RL policy has worse performance. This suggests that realistic grasping for a diverse set of objects (e.g. towel, knife, meat) remains a challenging robotics problem.
>
> Second, to address the concern of limited data for imitation learning experiments of bimanual pick and place, we collected 30 (instead of 5) demonstrations of 6500+ frames in total. The results show improved performance for episodes initialized further away from success. When trained with only 5 VR demonstrations, the agent completely failed when starting at T-7s. Now with 30 VR demos, it achieves a success rate of 15%. However, the performance doesn’t improve much for T-2 or T-4. This may indicate that the collected demonstrations are not sufficient for the complex task (for example, [1] uses 2500 demonstrations for bimanual lifting, and arguably bimanual pick and place is harder than that). After the rebuttal, we will attempt to collect more training data and perform data augmentation, as well as to use more advanced imitation learning algorithms. We would also like to note that the bimanual pick and place task is extremely challenging since it requires coordination between two hands and delicately balancing the loads between two hands. We didn’t completely solve the problem but provided a baseline and an example of how the VR interface can be leveraged to efficiently collect demonstrations for bimanual manipulation.
>
>
> Last but not least, we have also updated Sec. 6 and Appendix A.3 of the paper about our experimental evaluation accordingly.
>
> Thank you again for your time and feedback!
>
> - [1] Xie, Fan, et al. "Deep Imitation Learning for Bimanual Robotic Manipulation." Advances in Neural Information Processing Systems 33 (2020).
>
> Best,
>
> Paper293 Authors

---

### Official Review · Reviewer_kM4j · 2021-07-23

**Originality:** Very Good
**Technical Quality:** Good
**Clarity Of Presentation:** Very Good
**Impact:** 4

**Recommendation:**

Weak Accept: I recommend accepting the paper, but will not argue for my recommendation if the majority of other reviewers have a different opinion.

**Summary:**

This paper presents iGibson 2.0, an embodied robotics simulator that makes three novel contributions: (1) the introduction of object states such as temperature, cleanliness, and toggled states, (2) the inclusion of predicate logic functions that can be used to specify tasks and generate scenes, and (3) a virtual reality system to collect demonstrations. These contributions are aimed at increasing the complexity and diversity of embodied tasks while also facilitating human access to these tasks via improved user interfaces.

**Issues:**

* Explore the strengths and weaknesses of learning-based agents further. Either discuss or present empirical studies to answer the following questions: (1) what types of behaviors / tasks are especially difficult to solve (in addition to slicing)? (2) is partial observability an issue? ie. are there situations where the agent needs to keep track of the history of previous observations? (3) how sensitive is the agent to the camera viewpoint and the presence of distractor objects? (4) are there any modifications that should be made to learning-agents for dealing with object states?
* Consider re-doing the imitation learning experiment with a substantially higher number of demonstrations
* discuss the limitations of this work and future avenues of research

**Reviewer Expertise:**

Good: General knowledge of the area

**Strengths And Weaknesses:**

Strengths:
* The introduction is well-written and clear: it begins with a high-level discussion of the problem and then explains the solutions to address the problem
* The related work section is thorough: it lays out prior works and contrasts these works with the proposed approach
* The methods sections are thorough yet easy to understand: the figures especially complement the paper well
* The three technical contributions of the paper are all novel and useful to the embodied AI community, especially contribution #2: the ability to generate scenes with a large number of objects can enable the study of much more realistic tasks

Weaknesses:
* Some of the design choices — in particular modeling water as a set of discrete spheres — seem unrealistic, raising doubts on the possibility of sim-to-real transfer
* The RL experiments study relatively easy tasks: they consider relatively simple scenes, relatively short-horizon behaviors, dense rewards, and (presumably) full observability. Removing these assumptions is not explored in the experiments, making it difficult to understand the bottlenecks of learning-based agents in solving complex embodied tasks
* The imitation learning experiment is not informative: providing 5 demonstrations seems artificially low, are there constraints for having such few demonstrations?

**Summary Of Recommendation:**

Overall this paper is well-written, clear, and makes novel and useful technical contributions. The main consideration for improving the paper concerns the experiments — they study relatively simple tasks and the experimental analysis can go deeper. Refer to the list of issues for concrete suggestions.

**Update after rebuttal**: my recommendation stays the same. See post below for details.

---

> ### Author Response · Authors · 2021-08-27
> **Response to reviewer kM4j 1/2**
>
> Thanks for the constructive feedback and we provide responses as below:
>
> > The realism of some design choices
>
> We understand the reviewer’s concern: some approximations have been made in our simulator to achieve the speed necessary to train agents.
>
> We argue that our approximations are realistic enough to force AI agents to learn the necessary skills and motion to solve the tasks.
>
> For example, we do not simulate the most accurate thermodynamics for temperature change. Instead, we approximate with a system of heat sources and sinks that force the agent to learn to bring the objects to be cooked or frozen to be within the “functioning” radius of these sources/sinks (e.g. inside fridges/microwaves, near the heat source of stoves). The system also requires the agent to toggle on and/or close the door of these kitchen appliances for our “pseudo heat transfer” to happen.
>
> Similarly, while a full simulation of fluids would be more realistic, droplets “behave” to some extent similarly to liquid: they can be contained in and poured by container objects; they are created in special locations (e.g. sinks) and absorbed when in contact with towels and other `Soakable` objects.
>
> - In the case of slicing, the most accurate behavior would require modifying the meshes of the sliceable objects dynamically during simulation, which is a very costly procedure that forestalls real-time applications like our VR system. Our approximation, while clearly not exact, still forces the agent to learn to pick up a sharp tool, and use the right cutting edge with enough force on the surface of the sliceable object to cut it.
> - Based on this, we believe that the agents can learn strategies in iGibson 2.0 that, while may require adaptation to be transferred to the real world, are extremely useful to understand how to perform tasks in homes, such as using tools, cleaning surfaces, cooking food, etc.
> - During the development process, we strive to find a balance between the scalability and fidelity of the simulation. Some recent advances in physics simulators (Flex/PhysX and Taichi) have pushed the state-of-the-art performance of simulating liquid, soft cloth, and sliceable objects. We have experimented with some of those, and found that they still lacked the scalability to support large-scale, fully interactive scenes that iGibson has at the moment.
>
> The ideal solution would be to have a “heterogeneous” simulator that integrates different physics engines and apply them to where they perform the best, but this, apart from requiring a huge engineering effort, requires physics engines to “talk to each other”, affecting their states based on the computation of the others, which is currently not possible with most of them.
> In any case, we are exploring avenues to increase the realism of the simulation of the extended states, and adding additional ones such as Filled, Hung, Assembled, etc.
>
>
> > The RL experiments study relatively easy tasks (with assumptions. Exploring how these assumptions affect the results would be valuable.) (in the parenthesis we try to interpret what the reviewer mean.)
>
>
> According to a parallel submission BEHAVIOR (attached as supplementary material), these following factors contribute to the difficulty of household robotic tasks.
> - Long horizon: for relatively simple tasks that involve only one or two objects, we can achieve almost perfect success rates, but for those that involve many steps and objects, model-free RL algorithms don’t work anymore due to exploration challenges.
> - Physically based full-body control of the agent: We can achieve some success with an action space of magic actions or motion primitives. - But with physics simulation of full-body control, we achieve almost zero success rate.
> - Partial observability: Based on the experiments in a parallel submission BEHAVIOR, partial observability contributes to the difficulty of tasks, as the reviewer hypothesizes.
> - Reward shaping helps the agent to successfully achieve predicates. While using predicate as a sparse reward only, the agent doesn’t learn.

---

> ### Author Response · Authors · 2021-08-27
> **Response to reviewer kM4j 2/2**
>
> > The imitation learning experiment is not informative: providing 5 demonstrations seems artificially low, are there constraints for having such few demonstrations?
>
> This is a great point. Most imitation learning algorithms, and especially, straightforward ones such as behavioral cloning, require a larger amount of data to train robust policies. We will follow the recommendation, collect additional demonstrations, and retrain the policy. We will share our new results by the rebuttal deadline and revise the paper accordingly.
>
> > We added a section called “Limitations and Future Work” to our paper that we will also include here (see Appendix A.6 for the rebuttal and we will try to incorporate it into the main text in the final version).
>
> Although iGibson 2.0 has made several significant contributions towards simulating complex, everyday household tasks for robot learning, it is not without limitation. First of all, iGibson 2.0 doesn't support soft bodies / flexible material in a scalable way at the moment, due to the limitation of our underlying physics engine. This prevents us from simulating tasks like folding laundry and making bed in large, interactive scenes. Also, iGibson 2.0 doesn't support accurate human behavior modeling (other than goal-oriented navigation), and thus prevent us from simulating tasks that are inherently rich in human-robot interaction (e.g. elderly care). With the recent advancement of physics engines, and human behavior modeling and motion synthesis, we plan to overcome these limitations in the future. In addition, we also plan to support a more diverse set of extended object states (e.g. Filled, Hung, Assembled, etc) as well as bi-directional transition for some of our existing states (e.g. Soaked and Stained/Dusty), which can unlock even more household tasks. Finally, we plan to transfer mobile manipulation policies trained in iGibson 2.0 to the real world.

---

> ### Author Response · Authors · 2021-08-31
> **Follow-up response**
>
> Dear Reviewer,
>
> We would like to thank you again for your time and feedback on iGibson 2.0.
>
> Since we posted our response last Thursday, we have made additional progress towards addressing the raised concerns.
>
> First, to address the concern of applicability to robotics and realism of embodiment, we conducted additional RL experiments with the Fetch robot, with simplified grasping and without it. When trained with simplified grasping, the RL policy achieves perfect success for 4 out of the 5 tasks (similar to bimanual humanoid) while Slicing Fruit task remains unsolved (because it requires a precise alignment between the knife blade and the fruit). All the joint actuations of Fetch are realistically simulated in Bullet, except the grasping. When trained without simplified grasping, in which case the robot relies on the friction between the gripper fingers and the object to grasp it, the RL policy has worse performance. This suggests that realistic grasping for a diverse set of objects (e.g. towel, knife, meat) remains a challenging robotics problem.
>
> Second, to address the concern of limited data for imitation learning experiments of bimanual pick and place, we collected 30 (instead of 5) demonstrations of 6500+ frames in total. The results show improved performance for episodes initialized further away from success. When trained with only 5 VR demonstrations, the agent completely failed when starting at T-7s. Now with 30 VR demos, it achieves a success rate of 15%. However, the performance doesn’t improve much for T-2 or T-4. This may indicate that the collected demonstrations are not sufficient for the complex task (for example, [1] uses 2500 demonstrations for bimanual lifting, and arguably bimanual pick and place is harder than that). After the rebuttal, we will attempt to collect more training data and perform data augmentation, as well as to use more advanced imitation learning algorithms. We would also like to note that the bimanual pick and place task is extremely challenging since it requires coordination between two hands and delicately balancing the loads between two hands. We didn’t completely solve the problem but provided a baseline and an example of how the VR interface can be leveraged to efficiently collect demonstrations for bimanual manipulation.
>
>
> Last but not least, we have also updated Sec. 6 and Appendix A.3 of the paper about our experimental evaluation accordingly.
>
> Thank you again for your time and feedback!
>
> - [1] Xie, Fan, et al. "Deep Imitation Learning for Bimanual Robotic Manipulation." Advances in Neural Information Processing Systems 33 (2020).
>
> Best,
>
> Paper293 Authors

---

> > ### Comment · Reviewer_kM4j · 2021-09-03
> > **Reviewer response to authors**
> >
> > Thank you for taking the time to carefully consider my review and for making the corresponding revisions. After a careful study of your revisions and the discussions with the other reviewers, I am going to maintain my recommendation at weak accept. I think this is a good paper as is, and it perhaps should be somewhere between weak and strong accept. But there is still some room for improvement to consider:
> > * the number of demos collected for the bimanual task is now 30 instead of 5. That is certainly an improvement, but I think this is still on the low end. This suggests to me that perhaps collecting demonstrations for bimanual tasks with your VR interface may require expertise, and hence collecting a large number of demonstrations is challenging.
> > * Many reviewers raised concerns about the realism of the simulator for transferring policies to the real world. I think you outlined the reasons for why it Is difficult to accurately model the real world, but nevertheless the concern still stands. I found the new experiments on the simplified vs. non-simplified grasping to be helpful in demonstrating the challenge, but it was not really a surprising result for me.

---

### Official Review · Reviewer_WuTK · 2021-07-23

**Originality:** Good
**Technical Quality:** Very Good
**Clarity Of Presentation:** Excellent
**Impact:** 3

**Recommendation:**

Weak Accept: I recommend accepting the paper, but will not argue for my recommendation if the majority of other reviewers have a different opinion.

**Summary:**

This paper presents iGibson 2.0, an open-source simulation environment with three main features:
- a set of new physical properties such as temperature, wetness and cleanliness levels as well as unary and binary logical predicates such as "ToggledOn", "InsideOf", etc.
- generative functions to sample valid simulated states given the logical predicates
- a virtual reality interface to record human demonstrations for robot learning

**Issues:**

* Typos:
- Section 3: " They can also be contained in receptacles (e.g. cups) and poured later, leading to realistic behavior for the simulation of several household activities involving liquids, illustrated in Fig. 2a." --> The reference here should be to Fig. 2b.
- Figure 3b: "When it is toggled on and and its appearance changes accordingly." --> delete one "and"
- Figure 4: "Visualizations of densely populated ecological scenes"; " robot learning and ecological experience in virtual reality" --> is the desired term here really "ecological"?
- Section 6: "The results of this experiment are shown in Fig. 3b." --> The reference here should be to Fig. 5b.

**Reviewer Expertise:**

Good: General knowledge of the area

**Strengths And Weaknesses:**

The paper has two main strengths:
- The generative functions may enable researchers to easily generate diverse scenes with several objects to train and test robot learning algorithms.
- The virtual reality interface may be very useful to teach robots by demonstration or to initialize learning algorithms with human demonstrations.

Possible weaknesses of this work lie in the fact that the simulations are rather simplistic. When cooking, the food has only three stages: frozen, cooked or burnt; sliced objects are simply replaced by two halves; liquid is simulated as droplets. As a consequence, the utility of training learning algorithms in these simulated environments is questionable if the ultimate goal is to enable robots to solve tasks in the real world.

Some ideas for future work:
- an empty/full state for glasses, cups, bottles, etc.
- It might be interesting to consider simulating the possibility of making things dirty. For example, the preparation of food may make plates, cutlery and other objects dirty. One might want to work so that not too many things get dirty...



**Summary Of Recommendation:**

The generative capabilities of the proposed simulation environment may be helpful to researchers in need of creating diverse scenarios with several objects to train and test robot learning algorithms. The virtual reality interface presents a convenient way of giving demonstrations, especially for bimanual tasks.

The lack of realism of the simulations might prevent this environment from actually being used to train agents with the ultimate goal of deploying them in the real world.

---

> ### Author Response · Authors · 2021-08-27
> **Response to reviewer WuTK**
>
> Thanks for the constructive feedback and we provide responses as below:
>
> > The complexity/realism of simulation:
>
> We want to clarify that in our implementation, food has a continuous temperature value that is mapped by the logic predicates into three possible binary values: frozen, cooked and burnt. We understand the reviewer’s concern: some approximations have been made in our simulator to achieve the speed necessary to train agents.
>
> We argue that our approximations are realistic enough to force AI agents to learn the necessary skills and motion to solve the tasks.
>
> - For example, we do not simulate the most accurate thermodynamics for temperature change. Instead, we approximate with a system of heat sources and sinks that force the agent to learn to bring the objects to be cooked or frozen to be within the “functioning” radius of these sources/sinks (e.g. inside fridges/microwaves, near the heat source of stoves). The system also requires the agent to toggle on and/or close the door of these kitchen appliances for our “pseudo heat transfer” to happen.
> - Similarly, while a full simulation of fluids would be more realistic, droplets “behave” to some extent similarly to liquid: they can be contained in and poured by container objects; they are created in special locations (e.g. sinks) and absorbed when in contact with towels and other Soakable objects.
> - In the case of slicing, the most accurate behavior would require modifying the meshes of the sliceable objects dynamically during simulation, which is a very costly procedure that forestalls real-time applications like our VR system. Our approximation, while clearly not exact, still forces the agent to learn to pick up a sharp tool, and use the right cutting edge with enough force on the surface of the sliceable object to cut it.
>
> Based on this, we believe that the agents can learn strategies in iGibson 2.0 that, while may require adaptation to be transferred to the real world, are extremely useful to understand how to perform tasks in homes, such as using tools, cleaning surfaces, cooking food, etc.
>
> During the development process, we strive to find a balance between the scalability and fidelity of the simulation. Some recent advances in physics simulators (Flex/PhysX and Taichi) have pushed the state-of-the-art performance of simulating liquid, soft cloth, and sliceable objects. We have experimented with some of those, and found that they still lacked the scalability to support large-scale, fully interactive scenes that iGibson has at the moment.
>
> The ideal solution would be to have an “heterogeneous” simulator that integrates different physics engines and apply them to where they perform the best, but this, apart from requiring a huge engineering effort, requires physics engines to “talk to each other”, affecting their states based on the computation of the others, which is currently not possible with most of them.
> In any case, we are exploring avenues to increase the realism of the simulation of the extended states, and adding additional ones such as Filled, Hung, Assembled, etc
>
> > (would be good to support) an empty/full state for glasses, cups, bottles, etc.
>
> This is a really good idea! Many tasks in homes require understanding and modifying the empty/full state of containers. We have added this to the Future Work section (Appendix A.6). We will add this functionality in future versions. Thank you for the suggestion.
>
> > It might be interesting to consider simulating the possibility of making things dirty
>
> This is also a good idea. We currently have considered tasks where the agent needs to clean a dirty object, but it is a great observation that some tasks may create dirt, indirectly necessitating a cleaning task afterwards. The same applies to soakable: instead of allowing only one-direction change for these extended states (from dirty to clean, from dry to soaked), we could enable transitions in both directions, allowing the agent to make objects dirty or dry. This is a simple extension to our simulator that we are incorporating. We have added this to the Future Work section (Appendix A.6). We will add this functionality in future versions. Thank you for the suggestion.
>
> > Typos in the manuscript.  Usage of the term “ecological”.
>
> Thank you, we have fixed those typos.
>
> With respect to the term “ecological”, we use it here in the “Gibsonian” manner [1] indicating that it is a more naturalistic and realistic-looking environment than a plain, empty scene, where the behavior of the agent may be different.
>
> - [1] Gibson, James J. The ecological approach to visual perception: classic edition. Psychology Press, 2014.

---

> ### Author Response · Authors · 2021-08-31
> **Follow-up response**
>
> Dear Reviewer,
>
> We would like to thank you again for your time and feedback on iGibson 2.0.
>
> Since we posted our response last Thursday, we have made additional progress towards addressing the raised concerns.
>
> First, to address the concern of applicability to robotics and realism of embodiment, we conducted additional RL experiments with the Fetch robot, with simplified grasping and without it. When trained with simplified grasping, the RL policy achieves perfect success for 4 out of the 5 tasks (similar to bimanual humanoid) while Slicing Fruit task remains unsolved (because it requires a precise alignment between the knife blade and the fruit). All the joint actuations of Fetch are realistically simulated in Bullet, except the grasping. When trained without simplified grasping, in which case the robot relies on the friction between the gripper fingers and the object to grasp it, the RL policy has worse performance. This suggests that realistic grasping for a diverse set of objects (e.g. towel, knife, meat) remains a challenging robotics problem.
>
> Second, to address the concern of limited data for imitation learning experiments of bimanual pick and place, we collected 30 (instead of 5) demonstrations of 6500+ frames in total. The results show improved performance for episodes initialized further away from success. When trained with only 5 VR demonstrations, the agent completely failed when starting at T-7s. Now with 30 VR demos, it achieves a success rate of 15%. However, the performance doesn’t improve much for T-2 or T-4. This may indicate that the collected demonstrations are not sufficient for the complex task (for example, [1] uses 2500 demonstrations for bimanual lifting, and arguably bimanual pick and place is harder than that). After the rebuttal, we will attempt to collect more training data and perform data augmentation, as well as to use more advanced imitation learning algorithms. We would also like to note that the bimanual pick and place task is extremely challenging since it requires coordination between two hands and delicately balancing the loads between two hands. We didn’t completely solve the problem but provided a baseline and an example of how the VR interface can be leveraged to efficiently collect demonstrations for bimanual manipulation.
>
>
> Last but not least, we have also updated Sec. 6 and Appendix A.3 of the paper about our experimental evaluation accordingly.
>
> Thank you again for your time and feedback!
>
> - [1] Xie, Fan, et al. "Deep Imitation Learning for Bimanual Robotic Manipulation." Advances in Neural Information Processing Systems 33 (2020).
>
>
> Best,
>
> Paper293 Authors

---

> ### Comment · Reviewer_WuTK · 2021-09-01
> **Reviewer Response**
>
> Thank you for the detailed answers and for the additional work on the paper. My score remains the same.

---

### Official Review · Reviewer_zX2o · 2021-07-23

**Originality:** Very Good
**Technical Quality:** Excellent
**Clarity Of Presentation:** Excellent
**Impact:** 4

**Recommendation:**

Strong Accept: I recommend accepting the paper and will argue for my recommendation even if other reviewers hold a different opinion.

**Summary:**

This paper presents an open-source simulation environment iGibson 2.0 which supports a wide range of household tasks, extending the simulation from kinodynamics to other object states such as temperature etc. It provides a variety of functionalities, including logical states, generative scene initialization, and VR interface.



**Issues:**

I mainly have two questions about the experiment section:

- For experiments on iGibson tasks, how is the reward defined for tasks 1) to 5) ? Is it just a sparse reward function? Does it come with any reward shaping?

- What is the reason of not running rl algorithms on task 6)? It would be nice to see some justifications for this to understand the difficulty of task 6) compared to other tasks.


**Reviewer Expertise:**

Very good: Comprehensive knowledge of the area

**Strengths And Weaknesses:**

Strengths:
- An open-source efforts on simulation with very extensive functionalities. It provides a wide range of logical states description, which alleviates users' efforts to manually define a lot of logic states on their own. This could also speed up the research efforts on related tasks, making people focus more on the algorithmic designs.
- An interactive interface for easily collecting demonstration data.

Weakness:
- Seems that the relational states are limited to a pair of objects only. Would be nice to see an extension to this.
- The simulation seems to focus more on the embodied AI research. It is hard to see how to transfer a policy from simulation to the real world for robotics tasks.


**Summary Of Recommendation:**

This is  a simulation effort that will contribute to the community a lot. And the descriptions of the simulation is extensive.

---

> ### Author Response · Authors · 2021-08-27
> **Response to Reviewer zX2o**
>
>
>
> Thank you for your constructive review, we respond the questions as below:
>
> > Seems that the relational states are limited to a pair of objects only. Would be nice to see an extension to this.
>
> Good question! Currently we have focused on “unary” and “binary” predicates in first order logic, that is, predicates that take one object or a pair of objects as arguments. These are the most common ones used in logic when applied to physical tasks. However, as the reviewer pointed out, there could be predicates of higher arity, for example, `inBetween(o1, o2, o3)`. Our logic functionality allows future extensions to these types of predicates and we plan to do so as new activities require them.
>
> > How to transfer a policy from simulation to the real world for robotics tasks.
>
> Although our work has focused primarily on simulation, our goal is to use simulation as a proxy to develop solutions to real world robotics problems. All of our experiments are inherently robotic applications: pick and place, wiping, slicing, cooking, etc.
>
> To alleviate the concern of unrealistic embodiment, we will additionally include results of RL experiments with the Fetch robot embodiment and share our findings in Appendix A.3 by the rebuttal deadline. iGibson 2.0 also includes models of other real robots such as Freight, LoCoBot, Turtlebot, Quadcopter, etc.
>
> With respect to sim2real transferability, previous work [1] using the Fetch robot in iGibson has shown that the virtual sensor simulation is very realistic (see page 16) and such a platform has potential to develop solutions that transfer to the real world. Similar findings have been confirmed by other researchers using other types of robots for navigation tasks, e.g. four-wheeled robots [2] and legged robots [3].
>
> We also want to note that transferring mobile manipulation (MM) policies from simulation to the real world is an open research problem. While there has been successful transfer of navigation and stationary manipulation policies, the community is still a few steps away from transferring MM policies to the real world. Nonetheless, it is our plan to test some of our experiments on real robots in the near future now that we have regained access to the laboratory.
>
> > For experiments on iGibson tasks, how is the reward defined for tasks 1) to 5) ? Is it just a sparse reward function? Does it come with any reward shaping?”
>
> Task 1) to 5) have reward shaping. The reward function contains both a sparse, success reward, and a dense reward that encourages task progress. For example, task 2) has a reward shaping term that encourages the hand to approach the towel in the first stage and the towel to approach the faucet in the second stage. Task 3) also gives a partial reward for each stain particle that is cleaned/removed. The other tasks have similar reward shaping terms.
>
> > What is the reason of not running rl algorithms on task 6)? It would be nice to see some justification for this to understand the difficulty of task 6) compared to other tasks.”
>
> Task 6 is extremely complex as it involves coordinating not just one end-effector but two. This increases the dimensionality of the action space (12 degrees of freedom), which poses a hard-exploration problem for model-free reinforcement learning. While solutions exist (e.g., [4][5]), they go beyond the scope of our experimental evaluation. With this being said, we will attempt to run the same RL algorithm used for task 1-5 to task 6 and share our results later.
>
> ### References
> - [1] Xia, F., Li, C., Martín-Martín, R., Litany, O., Toshev, A., & Savarese, S. (2020). ReLMoGen: Leveraging motion generation in reinforcement learning for mobile manipulation. arXiv preprint arXiv:2008.07792.
> - [2] Meng, X., Xiang, Y., & Fox, D. (2021). Learning Composable Behavior Embeddings for Long-Horizon Visual Navigation. IEEE Robotics and Automation Letters, 6(2), 3128-3135.
> - [3] Li, T., Calandra, R., Pathak, D., Tian, Y., Meier, F., & Rai, A. (2021). Planning in learned latent action spaces for generalizable legged locomotion. IEEE Robotics and Automation Letters, 6(2), 2682-2689.
> - [4] Tung, Albert, et al. "Learning Multi-Arm Manipulation Through Collaborative Teleoperation." arXiv preprint arXiv:2012.06738 (2020).
> - [5] Ha, Huy, Jingxi Xu, and Shuran Song. "Learning a decentralized multi-arm motion planner." arXiv preprint arXiv:2011.02608 (2020).

---

> ### Author Response · Authors · 2021-08-31
> **Follow-up response**
>
> Dear Reviewer,
>
> We would like to thank you again for your time and feedback on iGibson 2.0.
>
> Since we posted our response last Thursday, we have made additional progress towards addressing the raised concerns.
>
> First, to address the concern of applicability to robotics and realism of embodiment, we conducted additional RL experiments with the Fetch robot, with simplified grasping and without it. When trained with simplified grasping, the RL policy achieves perfect success for 4 out of the 5 tasks (similar to bimanual humanoid) while Slicing Fruit task remains unsolved (because it requires a precise alignment between the knife blade and the fruit). All the joint actuations of Fetch are realistically simulated in Bullet, except the grasping. When trained without simplified grasping, in which case the robot relies on the friction between the gripper fingers and the object to grasp it, the RL policy has worse performance. This suggests that realistic grasping for a diverse set of objects (e.g. towel, knife, meat) remains a challenging robotics problem.
>
> Second, to address the concern of limited data for imitation learning experiments of bimanual pick and place, we collected 30 (instead of 5) demonstrations of 6500+ frames in total. The results show improved performance for episodes initialized further away from success. When trained with only 5 VR demonstrations, the agent completely failed when starting at T-7s. Now with 30 VR demos, it achieves a success rate of 15%. However, the performance doesn’t improve much for T-2 or T-4. This may indicate that the collected demonstrations are not sufficient for the complex task (for example, [1] uses 2500 demonstrations for bimanual lifting, and arguably bimanual pick and place is harder than that). After the rebuttal, we will attempt to collect more training data and perform data augmentation, as well as to use more advanced imitation learning algorithms. We would also like to note that the bimanual pick and place task is extremely challenging since it requires coordination between two hands and delicately balancing the loads between two hands. We didn’t completely solve the problem but provided a baseline and an example of how the VR interface can be leveraged to efficiently collect demonstrations for bimanual manipulation.
>
> Third, per reviewer zX2o’s request, we added an additional RL experiment for the bimanual pick and place. As we expected, the RL policy fails to achieve success due to the complicated two-hand coordination that this task requires. Despite a lot of effort, we couldn’t come up with an adequate reward function that can induce such collaborative behaviors within the rebuttal period.
>
> Last but not least, we have also updated Sec. 6 and Appendix A.3 of the paper about our experimental evaluation accordingly.
>
> Thank you again for your time and feedback!
>
> - [1] Xie, Fan, et al. "Deep Imitation Learning for Bimanual Robotic Manipulation." Advances in Neural Information Processing Systems 33 (2020).
>
> Best,
>
> Paper293 Authors

---

### Meta-Review · Area_Chair_mw8w · 2021-08-12

**Recommendation:** Accept (Poster)
**Confidence:** 4

**Metareview:**

All reviewers believe iGibson is an interesting simulator and they like the generative capabilities and tool for collecting demonstrations. The majority opinion seems positive. However, there are concerns about (a) simplistic representations (discrete states); (b) applicability to robotics — will sim2real work or how else do author envision this to be a CoRL paper; (c) detailed stud of what tasks would work and what would not. The authors should also redo IL experiments as kM4j asks for.

==
Post Rebuttal

The authors posted a strong response to the reviews and all reviews are now positive. AC appreciates all the work put in by the authors and recommends acceptance.

---

> ### Author Response · Authors · 2021-08-27
> **Response to Area Chair mw8w**
>
> We thank the reviewers and the area chair for their time and positive feedback on iGibson 2.0. Concretely, we are happy to hear that the reviewers value iGibson 2.0 as a way to “speed up the research efforts”, that its virtual reality interface is helpful for “easily collecting demonstration data” “to teach robots by demonstration”, and that “the three technical contributions of the paper are all novel and useful”.
>
> We acknowledge the concerns raised; we are trying to include as many of the suggested additional experiments as possible in the two-week rebuttal period. We have also revised our paper to provide additional information and clarification. Due to the page limit, most of the revisions are included in the Appendix (in red text). Note that the results of the additional experiments are still pending, and we will report them at the very end of the rebuttal period.
>
> In the following, we will reply to the reviewers one by one. Please let us know if you have any additional suggestions/questions.

---

> ### Author Response · Authors · 2021-08-31
> **Follow-up response to Area Chair**
>
> Dear Area Chair:
>
> We would like to thank you again for your time and feedback on iGibson 2.0.
>
> Since we posted our response last Thursday, we have made additional progress towards addressing the raised concerns.
>
> First, to address the concern of applicability to robotics and realism of embodiment, we conducted additional RL experiments with the Fetch robot, with simplified grasping and without it. When trained with simplified grasping, the RL policy achieves perfect success for 4 out of the 5 tasks (similar to bimanual humanoid) while Slicing Fruit task remains unsolved (because it requires a precise alignment between the knife blade and the fruit). All the joint actuations of Fetch are realistically simulated in Bullet, except the grasping. When trained without simplified grasping, in which case the robot relies on the friction between the gripper fingers and the object to grasp it, the RL policy has worse performance. This suggests that realistic grasping for a diverse set of objects (e.g. towel, knife, meat) remains a challenging robotics problem.
>
> Second, to address the concern of limited data for imitation learning experiments of bimanual pick and place, we collected 30 (instead of 5) demonstrations of 6500+ frames in total. The results show improved performance for episodes initialized further away from success. When trained with only 5 VR demonstrations, the agent completely failed when starting at T-7s. Now with 30 VR demos, it achieves a success rate of 15%. However, the performance doesn’t improve much for T-2 or T-4. This may indicate that the collected demonstrations are not sufficient for the complex task (for example, [1] uses 2500 demonstrations for bimanual lifting, and arguably bimanual pick and place is harder than that). After the rebuttal, we will attempt to collect more training data and perform data augmentation, as well as to use more advanced imitation learning algorithms. We would also like to note that the bimanual pick and place task is extremely challenging since it requires coordination between two hands and delicately balancing the loads between two hands. We didn’t completely solve the problem but provided a baseline and an example of how the VR interface can be leveraged to efficiently collect demonstrations for bimanual manipulation.
>
> Third, per reviewer zX2o’s request, we added an additional RL experiment for the bimanual pick and place. As we expected, the RL policy fails to achieve success due to the complicated two-hand coordination that this task requires. Despite a lot of effort, we couldn’t come up with an adequate reward function that can induce such collaborative behaviors within the rebuttal period.
>
> Last but not least, we have also updated Sec. 6 and Appendix A.3 of the paper about our experimental evaluation accordingly.
>
> Thank you again for your time and feedback!
>
> - [1] Xie, Fan, et al. "Deep Imitation Learning for Bimanual Robotic Manipulation." Advances in Neural Information Processing Systems 33 (2020).
>
> Best,
>
> Paper293 Authors

---

### Decision · Program_Chairs · 2021-09-13

**Decision:**

Accept (Poster)

**Comment:**

All reviewers believe iGibson is an interesting simulator and they like the generative capabilities and tool for collecting demonstrations. The majority opinion seems positive. However, there are concerns about (a) simplistic representations (discrete states); (b) applicability to robotics — will sim2real work or how else do author envision this to be a CoRL paper; (c) detailed stud of what tasks would work and what would not. The authors should also redo IL experiments as kM4j asks for.

==
Post Rebuttal

The authors posted a strong response to the reviews and all reviews are now positive. AC appreciates all the work put in by the authors and recommends acceptance.